# The Potential Advantages of Remimazolam for Awakening in Deep Brain Stimulation Surgery: A Retrospective Analysis of Cases

**DOI:** 10.3390/jcm14134724

**Published:** 2025-07-03

**Authors:** Sung-Hye Byun, Jinsong Yeo, Sou-Hyun Lee

**Affiliations:** 1Department of Anesthesiology and Pain Medicine, Kyungpook National University Chilgok Hospital, Kyungpook National University School of Medicine, Daegu 41404, Republic of Korea; stone0311@naver.com; 2Department of Anesthesiology and Pain Medicine, Kyungpook National University Hospital, Kyungpook National University School of Medicine, Daegu 41944, Republic of Korea

**Keywords:** anesthesia, deep brain stimulation, Flumazenil, neuroanesthesia, Remimazolam

## Abstract

**Background and Objectives:** Deep brain stimulation (DBS) requires sedation strategies that enable rapid and reliable awakening during intraoperative electrophysiological testing. Although propofol and dexmedetomidine are commonly used, their lack of pharmacological antagonists might delay recovery. In this retrospective case series, we assessed the effects of using remimazolam, a short-acting benzodiazepine that is reversible with flumazenil. No existing research has determined whether this may represent a clinically advantageous alternative. **Materials and Methods:** Six patients who underwent DBS surgery with monitored anesthetic care between May and August 2024 were included. Two patients received dexmedetomidine and propofol combined, whereas four received remimazolam for initial sedation. The time from sedation discontinuation to intraoperative electrophysiological examination, postoperative hospital stays, and perioperative complications were evaluated. **Results:** Patients who received remimazolam had shorter awakening intervals (median 17 min) compared to those who received dexmedetomidine and propofol (median 50 min), with a large effect size difference (Cliff’s delta −1.00). In all cases of remimazolam, patients were administered flumazenil to facilitate awakening, and transient hypertension requiring nicardipine was observed in some patients. Among the patients who underwent unilateral DBS, those who received remimazolam had shorter postoperative hospital stays (5–7 days) than the patient who received dexmedetomidine and propofol (9 days). No patient had complications. **Conclusions:** This small retrospective case series indicated that remimazolam, when reversed with flumazenil, was associated with rapid awakening compared with dexmedetomidine and propofol in patients undergoing DBS surgery. However, these findings require validation in larger prospective studies due to the small sample size.

## 1. Introduction

Deep brain stimulation (DBS) is a well-established neurosurgical intervention for managing movement disorders, including Parkinson’s disease, essential tremor, and dystonia [1]. This procedure involves the stereotactic implantation of stimulating electrodes through cranial burr holes guided by neuroimaging, with intraoperative microelectrode recordings facilitating accurate localization within the target nucleus [2]. Electrode placement is typically verified in the awake state to optimize precision [3].

Propofol and dexmedetomidine are frequently used for sedation during DBS; however, both agents are limited by the absence of a pharmacological antagonist, which may complicate excessive sedation management [4,5]. In contrast, the effects of remimazolam can be rapidly reversed with flumazenil, making it a potentially useful option in cases of excessive sedation [6]. This characteristic has led to its successful application in several awake craniotomies [7,8]. However, the use of remimazolam during awake DBS surgery has not been reported. Furthermore, patients who undergo DBS are generally older than those undergoing awake craniotomy (mean age 60.27 vs. 49.9 years) [9,10] and may benefit from remimazolam’s favorable emergence profile.

Considering the importance of timely awakening during intraoperative electrode localization and the increasing global volume of DBS surgeries [11], this study presents cases from our institution in which remimazolam was used for sedation during DBS procedures and comparative cases in which remimazolam was not used. Remimazolam’s pharmacodynamic profile includes a rapid onset, short context-sensitive half-time, and the ability to be antagonized with flumazenil [6]. These properties distinguish it from other sedatives used in monitored anesthesia care. Moreover, its use in neurosurgery is gaining attention, but high-level evidence for DBS-specific benefits remains limited.

Midazolam, similar to remimazolam, is a benzodiazepine; however, it was not considered due to its relatively slower onset and prolonged recovery time [12]. This case series highlights the practical advantages of remimazolam, particularly its reversibility with flumazenil, in facilitating a prompt transition to patient awakening during intraoperative electrophysiological testing.

In this study, we aimed to examine the use of remimazolam for sedation during DBS surgery and to evaluate its possible association with faster awakening compared with conventional sedatives. Furthermore, in this case series we seek to provide preliminary insights that may inform future research on optimal sedation strategies for DBS procedures.

## 2. Materials and Methods

### 2.1. Study Design

This retrospective case series was reviewed by the Institutional Review Board of Kyungpook National University Chilgok Hospital, which determined that formal approval was not required. Informed consent was obtained from all patients on the use of their de-identified data. All data were extracted from electronic medical records and fully de-identified to ensure patient anonymity.

### 2.2. Participants and Data

Patients who underwent DBS under monitored anesthesia between May and August 2024 were included.

The collected data variables included demographic information (age and sex), anesthetic records (operation, sedation duration, anesthetic drug dosages, vasopressor dosage, vasodilator dosage, and electrophysiology examination time), operative notes (surgical procedures), length of postoperative hospital stay, and the occurrence of complications such as stroke, reoperation, pneumonia, and angina.

### 2.3. Standard Deep Brain Stimulation Procedure Under Monitored Anesthetic Care

The patients underwent magnetic resonance imaging for anatomical localization of the target structures; the Leksell frame was placed before entering the operating room. Upon arrival in the operating room, standard monitoring devices including a pulse oximeter and a three-lead electrocardiogram were applied. Transnasal high-flow oxygen therapy (CO-FLO III; Chinoomed (Bucheon, Republic of Korea)) was initiated at a flow rate of 10 L/min. For continuous blood pressure monitoring, a 20-gauge radial artery catheter was inserted into the left radial artery following local infiltration with lidocaine.

The surgical procedure was performed as follows [13]: Once adequate sedation was confirmed, the neurosurgeon infiltrated lidocaine locally into the scalp, followed by a burr hole creation. Under fluoroscopic guidance, a microelectrode was advanced into the target region. The patient was awakened for the intraoperative electrophysiological examination. After confirming the optimal electrode position, a macroelectrode was inserted, and a second electrophysiological assessment was conducted. The surgical site was closed. Subsequently, general anesthesia was induced, and an implantable pulse generator was placed in the subclavicular region. Following completion of the procedure and emergence from anesthesia, the patient was transferred from the operating room.

Sedation was initiated using one of the two following methods, which was selected according to the anesthesiologist’s preference: (1) a combination of dexmedetomidine (all patients received a loading dose up to 1.0 µg/kg for 10 min followed by continuous infusion at 0.2 µg/kg/h) and propofol (target effect-site concentration of 1.5 µg/mL using the Schnider model [14]), or (2) remimazolam (all patients received a bolus of up to 0.1 mg/kg followed by continuous infusion at 0.5 mg/kg/h) alone. Remifentanil was administered to all patients, regardless of the sedation regimen. The sedative agents were discontinued at the initiation of the burr hole procedure in all patients. All sedative agents were administered using a Perfusor Space Syringe Pump (B. Braun Medical Inc. (Bethlehem, PA, USA)).

### 2.4. Accessing Patient Awakening

Depth-of-anesthesia monitors such as the Bispectral Index (BIS) or Patient State Index (PSI) could not be used following interference from intraoperative neurophysiological equipment (Leksell frame). Therefore, awakening was assessed using the Modified Observer’s Assessment of Alertness/Sedation Scale (MOAA/S), which was well correlated with BIS and PSI in previous studies [15,16]. Awakening was defined as the patient responding to their name spoken in a normal tone and corresponding to a MOAA/S score of 5. In patients who received remimazolam, flumazenil 0.2 mg was administered intravenously at 10 min after discontinuation of remimazolam infusion if the MOAA/S score remained below 5. No patients in the dexmedetomidine–propofol group received flumazenil. The interval between the discontinuation of anesthetic agents and initiation of the intraoperative electrophysiological examination was defined as the awakening time. For patients who underwent bilateral procedures, the awakening time was defined based on the first side.

### 2.5. Statistical Analysis

Awakening times between the two sedation methods were compared using the Mann–Whitney U test and are presented as medians with interquartile ranges. Postoperative hospital stay is compared between patients who underwent bilateral vs. unilateral DBS using the Mann–Whitney U test and is presented as medians with interquartile ranges. The effect sizes were calculated using Cliff’s delta [17]. All statistical analyses were performed using RStudio version 4.5.0 (RStudio (Boston, MA, USA)).

## 3. Results

Six patients underwent DBS surgery under monitored anesthesia between May and August 2024. Two patients received dexmedetomidine and propofol, and four received remimazolam for sedation. A summary of the cases is presented in Table 1.

The patients in cases 1 and 2, who received dexmedetomidine and propofol, had awakening times of 40 and 60 min, respectively. Cases 3, 4, 5, and 6, who received remimazolam, had awakening intervals of 15, 25, 15, and 19 min, respectively. A clinically significant large effect size (Cliff’s delta = −1.00, 95% confidence interval (CI) −1.00 to −0.02) was observed between the two sedation methods despite the difference not being statistically significant (*p* = 0.133) (Table 2).

Postoperative hospital stays were 15, 21, and 10 days for the three patients who underwent bilateral DBS procedures, and 9, 5, and 7 days for those who underwent unilateral procedures. The analysis revealed a clinically significant large effect size (Cliff’s delta = −1.00, 95% CI −1.00 to −0.14) despite the difference in postoperative hospital stay between unilateral and bilateral procedures not being statistically significant (*p* = 0.100) (Table 3). Among the patients who underwent unilateral surgery, the patient in case 1 (dexmedetomidine and propofol) had a postoperative hospital stay of 9 days, whereas those in cases 3 and 6 (remimazolam) had stays of 5 and 7 days, respectively. No patient had documented complications during hospitalization.

In all of the patients who received remimazolam (cases 3, 4, 5, and 6), flumazenil (0.2 mg) was administered before the intraoperative electrophysiological examination due to the persistent sedation observed 10 min after the discontinuation of remimazolam infusion. For patients who underwent bilateral DBS surgery (cases 4 and 5), flumazenil was administered more than once. Patients who received flumazenil required nicardipine to manage their elevated blood pressure after flumazenil administration (Figure 1).

## 4. Discussion

In this case series, patients who received remimazolam for sedation awakened faster than those who received dexmedetomidine and propofol, showing clinically significant differences. For patients administered remimazolam, flumazenil was required to achieve full wakefulness, and nicardipine was required for the transiently elevated blood pressure following flumazenil administration. Patients who underwent unilateral DBS surgery and received remimazolam had shorter postoperative hospital stays than those who received the other drugs (5 and 7 days vs. 9 days). No complications were observed in any of the patients.

The combination of propofol and dexmedetomidine offers several clinical advantages over propofol alone. The time to achieve sedation is comparable with that of propofol alone; however, the incidence of adverse effects such as oxygen desaturation and nausea is reduced when the two agents are combined [18,19]. In one study, the mean time to eye opening was 20 min after the administration of propofol, remifentanil, and dexmedetomidine, without a loading dose, in patients with a mean age of 74.5 years [20]. In this case series, cases 1 and 2 required 40 and 60 min, respectively, to achieve full awakening. This discrepancy may be attributed to differences in the definition of awakening endpoints, specifically between eye opening and attaining full wakefulness, which includes the ability to follow verbal commands. In addition, the use of a dexmedetomidine loading dose in the present cases may have contributed to the prolonged awakening time. Another study involving patients with a mean age of 72 years reported a mean awakening time of approximately 12 min when dexmedetomidine was administered at a loading dose of 0.5 µg/kg for sedation, and in the study propofol was used for maintenance [21]. In that study, dexmedetomidine was not continued as a maintenance infusion, which may explain the shorter awakening time compared to this present study. To our knowledge, no study has evaluated the combination of propofol and dexmedetomidine in the context of intracranial surgery, thereby making direct comparisons difficult. However, the shorter awakening times observed in the cited studies [20,21] may be attributed to the use of BIS-guided sedation protocols and the procedures not being neurosurgical, compared with the present study where objective depth monitoring was not feasible.

Chae et al. [22] recommended a remimazolam bolus dose of 0.19–0.33 mg/kg for patients aged 60–80 years and 0.14–0.19 mg/kg for those older than 80 years undergoing general anesthesia. In this case series, the bolus dose was reduced to ≤0.1 mg/kg as the intended goal was sedation rather than general anesthesia. Patients who received remimazolam during anesthesia induction did not wake up after 10 min; therefore, flumazenil administration was required. Although remimazolam is known for its short elimination half-life and fast awakening [23], time to full awakening following 1 h of infusion may extend to 2 h in older patients [24]. Therefore, in the present cases, flumazenil was required to enable the timely initiation of the intraoperative electrophysiological examination for accurate electrode localization. Previous studies have reported that the use of flumazenil can significantly contribute to rapid awakening after remimazolam administration [25,26].

In this case series, postoperative hospital stay was shorter for patients who received remimazolam (cases 3 and 6) compared to those who received dexmedetomidine and propofol (case 1). However, despite the association between remimazolam and flumazenil in facilitating faster awakening, previous studies did not demonstrate a reduction in postoperative hospital stay [24]. As this study involved only six patients, further prospective research with a larger sample size is warranted. Among the patients who underwent bilateral DBS procedures, no notable difference in postoperative hospital stay was observed between the patients who received remimazolam (cases 4 and 5) and those who received dexmedetomidine and propofol (case 2). Given that a longer operative duration is a major factor associated with prolonged hospitalization [27], it is unlikely that the use of remimazolam alone is a primary determinant of reduced postoperative hospital stay.

In this study, hypertension was observed after flumazenil administration, which was defined as an increase of 20% or more from pre-anesthesia blood pressure [28]. Nicardipine was administered if the elevated blood pressure persisted for more than 5 min. This response may have been related to either the resolution of the sedative effects of remimazolam or an increase in sympathetic nervous system activity [29]. Previous findings suggested that flumazenil does not significantly alter cerebral blood flow or metabolism in patients in neurosurgery [30]; however, it should be used cautiously as hypertensive responses may pose a risk for intracranial hemorrhage, myocardial injury, or arrhythmia [31]. Although the association between flumazenil and seizure occurrence remains controversial [32,33], dose reduction is generally recommended because of its potential risks [34]. Accordingly, a 0.2 mg dose of flumazenil was administered to patients who received remimazolam for sedation in this study.

This case series has some limitations. First, sedative drug administration was not guided by objective anesthetic depth monitoring devices, such as the PSI or BIS. This was due to an overlap with intraoperative neurophysiological monitoring equipment. Consequently, a potential bias may exist in the assessment of the awakening time. However, patient awakening was evaluated using the MOAA/S to minimize this bias, which correlates appropriately with PSI and BIS. To enable accurate comparisons, future studies should incorporate standardized monitoring of the sedation depth. Second, the sedation regimens in this case series were not randomized and were distributed unequally between groups. Therefore, to address this limitation, the effect size was calculated to supplement the interpretation of group differences in awakening time between the two sedation methods.

The rapid awakening observed following flumazenil administration in the patients who received remimazolam, compared with those who received dexmedetomidine and propofol, supports the potential utility of remimazolam for sedation in patients undergoing DBS surgery. Although the findings of this case series are not generalizable due to the small sample size, to our knowledge this is the first study to compare awakening times between remimazolam and other sedative agents in the context of DBS procedures. Future studies should include prospective, randomized comparisons between remimazolam–flumazenil and dexmedetomidine–propofol sedation regimens. The use of electroencephalographic (EEG) monitoring with shielded equipment or time-synchronized offline analysis may enable the reintroduction of objective sedation depth assessment, even in the presence of neurophysiological monitoring systems. Therefore, this case series offers valuable preliminary insights for future research.

## 5. Conclusions

In conclusion, this pilot case series suggested that remimazolam, when reversed with flumazenil, may facilitate faster awakening compared with dexmedetomidine and propofol in patients undergoing DBS surgery. However, definitive conclusions cannot be drawn following the small sample size, retrospective design, and absence of objective anesthetic depth monitoring. These may introduce biases in the awakening time assessments. Despite the results not being generalizable, to our knowledge this is the first study to compare awakening profiles between remimazolam and other sedative agents in regard to DBS procedures. These preliminary findings provide valuable insights and highlight the necessity for larger, prospective studies to establish evidence-based sedation protocols for DBS surgery.

## Figures and Tables

**Figure 1 jcm-14-04724-f001:**
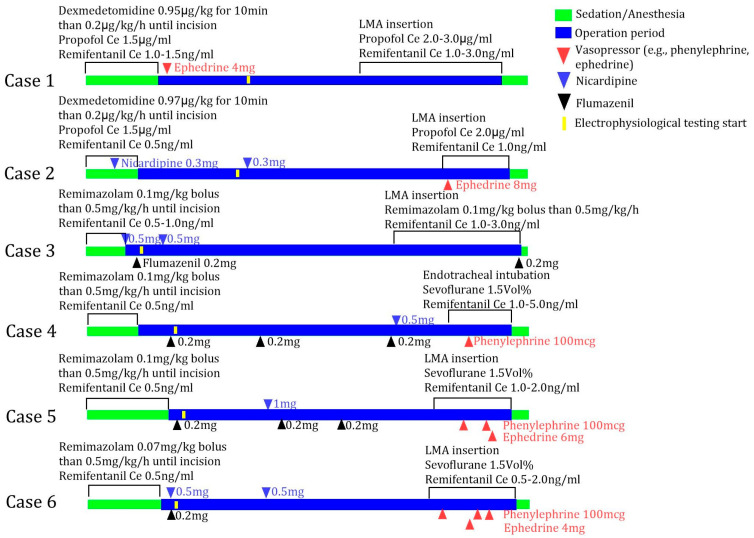
Timeline of sedation, anesthesia, and intraoperative drug administration for six cases. Propofol and remifentanil were administered using target-controlled infusion (Schnider model for propofol and Minto model for remifentanil). Ce, effect-site concentration; LMA, laryngeal mask airway.

**Table 1 jcm-14-04724-t001:** Case summaries for awake DBS surgery.

Cases	Sedation Drug	Operation Site	Awakening Time (min)	Operation Duration (min)	Postoperative Hospital Stay (day)	Complication
Case 1 (F/70 years)	Propofol and dexmedetomidine	Left	40	190	9	None
Case 2 (M/78 years)	Propofol and dexmedetomidine	Bilateral	60	306	15	None
Case 3 (F/73 years)	Remimazolam	Left	15	314	5	None
Case 4 (F/70 years)	Remimazolam	Bilateral	25	312	21	None
Case 5 (F/62 years)	Remimazolam	Bilateral	15	267	10	None
Case 6 (F/84 years)	Remimazolam	Left	19	229	7	None

Awakening time refers to the minutes from the discontinuation of the anesthetic drug infusion to patient awakening and the initiation of the electrophysiological examination. For bilateral surgeries, the time for the first side is reported. Complication includes stroke/reoperation/pneumonia/angina. DBS, deep brain stimulation; F, female; M, male.

**Table 2 jcm-14-04724-t002:** Awakening time comparison between two sedation methods.

	Remimazolam (*n* = 4)	Propofol, Dexmedetomidine (*n* = 2)	*p*-Value *	Effect Size ^#^ (95% CI)
Awakening time (min)	17 (15–23)	50 (40–60)	0.133	−1.00 (−1.00, −0.02)

Awakening time refers to the minutes from the discontinuation of the anesthetic drug infusion to patient awakening and the initiation of the electrophysiological examination. For bilateral surgeries, the time for the first side is reported. CI, confidence interval. ***** Mann–Whitney U test. ^#^ Cliff’s delta (absolute value < 0.33, small; 0.33–0.474, medium; >0.474, large).

**Table 3 jcm-14-04724-t003:** Postoperative hospital stay between operation site.

	Unilateral (*n* = 3)	Bilateral (*n* = 3)	*p*-Value *	Effect Size ^#^ (95% CI)
Postoperative hospital stay (day)	7 (5–9)	15 (10–21)	0.100	−1.00 (−1.00, −0.14)

CI, confidence interval. ***** Mann–Whitney U test. ^#^ Cliff’s delta (absolute value < 0.33, small; 0.33–0.474, medium; >0.474, large).

## Data Availability

The datasets generated and/or analyzed in the current study are available from the corresponding author upon reasonable request.

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
