# Peer review of "The Potential Advantages of Remimazolam for Awakening in Deep Brain Stimulation Surgery: A Retrospective Analysis of Cases"

_jcm, 2025, doi:10.3390/jcm14134724_

Round 1
Reviewer 1 Report
Comments and Suggestions for Authors
This case series describes the novel use of remimazolam (with flumazenil reversal) for sedation during awake DBS surgery in 6 patients, comparing awakening times to a dexmedetomidine-propofol regimen. As a descriptive clinical report, it provides valuable preliminary insights into a potentially advantageous sedation strategy. Within the limitations of the case series design, significant revisions to the manuscript were required to improve methodological clarity and contextify the findings.
Major Comments​​
- The series includes ​​only 6 patients​​ (4 remimazolam, 2 dexmedetomidine/propofol), limiting the validity of comparisons and conclusions.
- Cases are heterogeneous (unilateral vs. bilateral DBS, variable surgical durations), which complicates interpretation of outcomes like awakening time and hospital stay.
- Lack of explicit acknowledgment that sedation regimens were non-randomized and unequally allocated (2 vs. 4 patients).
- Incomplete quantification of flumazenil-induced hypertension (e.g., BP values, duration, intervention thresholds).
- Lack of Sedation Depth Monitoring: Objective measures (e.g., Bispectral Index) were not used, potentially introducing bias in awakening time assessment .
- Sedation Protocols: The dosing regimens for remimazolam (0.1 mg/kg bolus + 0.5 mg/kg/h infusion) and dexmedetomidine/propofol are not standardized across cases. For example, dexmedetomidine loading doses were used in some cases but not specified in others.
- Subjective "awakening time" definition risks measurement bias.
- The authors attribute shorter awakening times to remimazolam’s pharmacokinetics but fail to consider confounding factors (e.g., depth of sedation, surgical stimulation during awakening).
Minor Comments​​
​​Figure 1: Improve labeling of timelines (e.g., annotate key events like "electrophysiological testing start").
​​References: Ensure all citations are up-to-date and relevant (e.g., recent trials comparing sedatives in DBS).
The reference numbers are in duplicate.
Reviewer 2 Report
Comments and Suggestions for Authors
This manuscript presents a retrospective case series comparing remimazolam with dexmedetomidine-propofol for sedation during awake DBS. The study highlights remimazolam’s potential advantages, particularly its reversibility with flumazenil, in achieving faster awakening.
While the topic is clinically relevant, the study has several weaknesses that must be addressed before publication.
I provide detailed and constructive comments to help the authors improve their manuscript.
- The abstract needs improvements. Emphasize the exploratory nature of the findings (e.g., "This small retrospective case series suggests…").
- Clearly mention the study’s limitations (e.g., "Due to the small sample size, these findings require validation in larger prospective studies.").
- The introduction doesn’t justify why remimazolam (a relatively new drug) was chosen over other reversible sedatives (e.g., midazolam).
- Discuss why remimazolam was selected over other benzodiazepines (e.g., pharmacokinetic advantages in elderly patients).
- The introduction ends abruptly; a stronger transition to the study’s aim is needed.
- Methods: there’s a lack of standardized depth monitoring. No BIS or PSI monitoring, which is critical for sedation studies.
- Justify the lack of depth monitoring (e.g., interference with neurophysiological equipment).
- Remimazolam bolus doses varied (0.07–0.2 mg/kg), which could confound results.
- No statistical analysis: Descriptive statistics alone are insufficient.
- If possible, perform basic statistical tests (e.g., Mann-Whitney U test for awakening times).
- Results: Confounding factors: Bilateral vs. unilateral DBS cases are mixed, making comparisons unreliable.
- Separate bilateral and unilateral cases in the analysis.
- No statistical comparison: ideas of "shorter awakening times" are not supported by statistical testing.
- Transient hypertension is not quantified (e.g., BP values, duration).
- Quantify blood pressure changes after flumazenil.
- Discussion should be more cautious. Small case series cannot support claims like "remimazolam may be more advantageous."
- Cites studies where dexmedetomidine-propofol had faster awakening (20 min vs. 40–60 min here), but does not adequately explain the discrepancy.
- Discuss why awakening times differed from prior studies (e.g., loading dose of dexmedetomidine).
- Dicussion downplays the significance of hypertension post-flumazenil, which could be clinically important.
- Address whether hypertension post-flumazenil could impact DBS safety (e.g., intracranial hemorrhage risk).
- Conclusion exaggerates results (e.g., "remimazolam may be more advantageous") without sufficient evidence.
- Mention that this is a pilot study (e.g., "This small case series provides preliminary evidence… but larger trials are needed.").
- Conclusion doesn’t mention the study’s limitations.
- Clearly mention that no definitive conclusions can be drawn due to the sample size and design.
- "Awakening time" is defined but later referred to as "recovery time" in some places.
Author Response
Please see the attachmen

Round 2
Reviewer 1 Report
Comments and Suggestions for Authors
The author has made detailed revisions to the article. I have no further suggestions.
Reviewer 2 Report
Comments and Suggestions for Authors
The authors have addressed my comments well.